# Re-Evaluation of the Prevalence of Permanent Congenital Hypothyroidism in Niigata, Japan: A Retrospective Study

**DOI:** 10.3390/ijns7020027

**Published:** 2021-05-28

**Authors:** Keisuke Nagasaki, Hidetoshi Sato, Sunao Sasaki, Hiromi Nyuzuki, Nao Shibata, Kentaro Sawano, Shota Hiroshima, Tadashi Asami

**Affiliations:** 1Department of Homeostatic Regulation and Development, Division of Pediatrics, Niigata University Graduate School of Medical and Dental Sciences, Niigata 951-8510, Japan; totsutotsu0118@gmail.com (H.S.); sunaoenari@gmail.com (S.S.); nyuzuki@med.niigata-u.ac.jp (H.N.); shibata8400@gmail.com (N.S.); sawano@med.niigata-u.ac.jp (K.S.); sho980522@gmail.com (S.H.); 2Department of Pediatrics, Nagaoka Institute for Severely Handicapped Children, Nagaoka 940-2135, Japan; tasami@n-seiryo.ac.jp

**Keywords:** congenital hypothyroidism, newborn screening, Japan, re-evaluations, prevalence

## Abstract

Although newborn screening (NBS) for congenital hypothyroidism (CH) in Japan started more than 40 years ago, the prevalence of CH remains unclear. Prevalence estimations among NBS-positive CH individuals include those with transient hypothyroidism and transient hyperthyrotropinemia, and re-evaluation with increasing age is necessary to clarify the actual incidence. Thus, we re-evaluated the incidence of permanent CH. Of the 106,114 patients who underwent NBS in the Niigata Prefecture, Japan, between April 2002 and March 2006, 116 were examined further due to high thyroid-stimulating hormone levels (>8 mIU/L) and were included in the study. We retrospectively evaluated their levothyroxine sodium (LT4) replacement therapy status from the first visit to 15 years of age. Of the 116 NBS-positive patients, 105 (91%) were initially examined in our department. Of these, 72 (69%) started LT4 replacement therapy on the first visit. Subsequently, 27 patients continued LT4 replacement until 15 years of age after multiple re-evaluations. The prevalence of permanent CH in the Niigata Prefecture during this period was 1 in 2500–3500 children. Ultimately, 62.5% of patients on LT4 replacement discontinued treatment by 15 years of age. This is the first study to clarify the true prevalence of permanent CH in Japan.

## 1. Introduction

In Japan, newborn screening (NBS) for congenital hypothyroidism (CH) was initiated in 1979. The prevalence of CH in Japan was initially estimated at 1 in 7400 newborns prior to the commencement of the NBS [1]. National survey results after starting NBS indicated that the prevalence of CH in Japan was 1 in 1600–2500 children since the 2000s [1,2]. CH detected by NBS has been reported in various countries worldwide, but the incidence varies from 1 in 1000 to 1 in 6000 children [3]. Recently, reports have indicated that the prevalence of CH detected by NBS is increasing due to lower thyroid-stimulating hormone (TSH) cut-off values, racial composition changes, and an increase in the number of preterm or low-birthweight infants [3,4,5].

NBS-positive individuals present with transient hypothyroidism and transient hyperthyrotropinemia [6], and re-evaluation with increasing age is necessary to differentiate between these conditions and determine the actual incidence of CH. Permanent CH prevalence, excluding transient hypothyroidism, has not been clarified in Japan, which prompted our retrospective re-evaluation. To clarify the true prevalence of permanent CH in Niigata, Japan, we re-evaluated the patients who were NBS-positive for CH to determine the permanent CH prevalence based on the levothyroxine sodium (LT4) replacement status.

## 2. Materials and Methods

This was a single-institution retrospective cohort study. We retrospectively reviewed the LT4 replacement therapy status from the first visit after birth to 15 years of age. In this study, patients on LT4 replacement were defined as CH patients.

### 2.1. NBS Method in the Niigata Prefecture

Blood samples were collected on filter paper within the first 4 to 7 postnatal days, and the TSH level in the filter paper sample was measured using an enzyme-linked immunosorbent assay (TSH: Enzaplate N-TSH, Bayer Co., Tokyo, Japan). All CH screening tests were centralized at the Niigata Health Laboratory Center.

If the initial TSH level was between 8 and 30 mU/L, a second specimen was evaluated. If the TSH level in the second specimen was also greater than 8 mU/L, a confirmatory test was performed within 30 days of birth at the patient’s medical institution. If the initial TSH level was more than 30 mU/L, a confirmatory test was performed within 14 days of birth at the patient’s medical institution. Serum-free T4 (FT4), free T3 (FT3), and TSH levels were measured, and the thyroid morphology was evaluated by ultrasonography at the patient’s medical institution.

The included patients comprised 116 newborns who tested positive for high TSH levels among the 106,114 newborns who underwent NBS in the Niigata Prefecture, Japan, between April 2002 and March 2006. Patients who had been initially examined at other hospitals were excluded.

### 2.2. Re-Evaluations at Ages 2–4 Years

Patients with a eutopic thyroid gland who underwent LT4 replacement therapy and remained euthyroid without LT4 dose increments after 12 months of age were re-evaluated by discontinuing LT4 for 4 weeks and performing thyroid function tests.

### 2.3. Etiological Diagnosis Determination for CH after 5 Years of Age

The methods are detailed in previous studies [7]. To summarize, after discontinuing LT4 replacement therapy, several tests (such as thyroid function test, thyrotropin-releasing hormone stimulation, ^123^I scintigraphy and radioactive iodine uptake (RAIU), saliva-to-plasma radioiodine ratio, and perchlorate discharge (if the RAIU was 20% or more)) and thyroid ultrasonography were performed.

### 2.4. Re-Evaluations at Final Height

Patients with a eutopic thyroid gland who underwent LT4 replacement therapy and remained euthyroid after achieving their final height were re-evaluated by discontinuing LT4 for 4 weeks to confirm thyroid function.

### 2.5. Criteria for Beginning and Discontinuing LT4 Replacement Therapy

The criteria for beginning LT4 were a serum TSH level of 10–15 mU/L or higher at the time of the initial visit, persistent TSH level of ≥10 mU/L after the age of 3–6 months, or a persistent TSH level of ≥5 mU/L after the age of 1 year. The discontinuation criterion was a serum TSH level of <5 mU/L without LT4 replacement therapy, which was restarted if the TSH level remained above the 5–10 mU/L range.

### 2.6. Primary and Secondary Outcomes

The primary outcome was the prevalence of permanent CH detected by NBS, and the secondary outcome was the prevalence of transient CH among patients with CH who received LT4 replacement. This study was approved by the Niigata University Ethics Committee. We have published information related to the content of the research on the hospital’s homepage. The patients and their parents were informed of their right to refuse access to their medical records for use in the study.

## 3. Results

The background characteristics of the subjects are listed in Table 1. Fifteen percent of the NBS-positive infants had a low birthweight. Of the 116 NBS-positive subjects with high TSH levels, 105 (91%) were evaluated at our hospital (Figure 1). Therefore, this study is based on a population base of 106,114 × 91% (i.e., 96,000 newborns). The LT4 replacement status for each age group is shown in Figure 2. Of these, 73 patients (69%) were initiated on LT4 at their first visit, while 32 (31%) were left initially untreated; 10 of the latter had persistent mildly elevated TSH levels and were initiated on LT4 by the age of 1 year. Thus, 73 out of 87 patients (84%) were treated with LT4 for 2 years, excluding those who were transferred or those for whom the follow-up had ended.

Among patients aged between 2 and 5 years, 55 patients were re-evaluated and 16 discontinued LT4 replacement therapy. Consequently, 57 patients were on LT4 replacement therapy at the age of 5 years; of these, 52 patients were diagnosed etiologically with CH at the age of 5–7 years. LT4 replacement was discontinued in 24 patients, and 33 patients were continued on LT4 after the CH etiological diagnosis.

At the re-evaluation conducted after reaching final height, six patients discontinued LT4 replacement therapy, and at 15 years of age, 27 of the 79 patients (34%) who were followed up were receiving LT4 replacement therapy.

### 3.1. Permanent CH Prevalence

In addition to the 27 patients receiving LT4 replacement at 15 years of age, 10 patients were transferred or died while on LT4 replacement therapy. Thus, the number of patients with permanent CH from April 2002 to March 2006 ranged from 27 to 37, and the permanent CH prevalence was 1 in 2500–3500 children.

### 3.2. Transient CH Prevalence

Of the 74 patients who received LT4 replacement at 1 year of age, LT4 was discontinued in 46 patients by the age of 15 years, suggesting transient CH. There were 13 NBS-positive infants with an elevated TSH level and a birthweight of less than 2500 g who were examined at our hospital; 11 were initiated on LT4 replacement therapy, 3 were transferred to other hospitals, and 8 discontinued LT4 by the age of 15 years. Thus, the number of patients with transient CH or transient hyperthyrotropinemia ranged from 46 to 56, and the transient CH or transient hyperthyrotropinemia prevalence was 1 in 1700–2100 children.

## 4. Discussion

In this study, the re-evaluated prevalence of permanent CH was 1 in 2500–3500 children. Approximately 60% of the patients who received LT4 replacement therapy had transient CH or transient hyperthyrotropinemia and discontinued LT4 replacement therapy.

The CH prevalence reported worldwide is likely to include patients with transient hypothyroidism. In our study, when a patient with CH was defined as a patient on LT4 replacement therapy, the CH prevalence was approximately 1 in 1300 children at the time of the first visit after birth, 1 in 1200 children at 1 year of age, and 1 in 1500 children at 5 years of age. The prevalence of CH at 1 year of age increased from the first visit due to the inclusion of patients who started LT4 replacement in early infancy without LT4 replacement at the initial diagnosis, i.e., persistent mild hyperthyrotropinemia. Therefore, when discussing CH prevalence, it is difficult to compare without considering the timing (i.e., age) of the incidences.

Reports indicate that the frequency of transient CH has increased, likely because of lower TSH level cut-off values. The incidence of transient CH in North America is approximately 5% to 10% of the NBS-positive children with CH [8]; however, recent reports indicate that 40% to 53% of the NBS-positive children with CH actually have transient CH [9,10]. Only one-third of patients with CH and eutopic thyroid gland needed to continue LT4 replacement after re-evaluation at the age of 3 to 6 [11]. In our study, approximately 60% of the 74 patients on LT4 replacement therapy at the age of 1 year discontinued it by the age of 15 years. Thus, the TSH cut-off value of 8 mU/L does include a certain number of patients with transient hypothyroidism. Even if LT4 replacement cannot be discontinued by 5 years of age, some patients may be able to discontinue LT4 thereafter and should be re-evaluated for transient CH at the appropriate period.

Increasing numbers of low birthweight and preterm infants may also be associated with CH prevalence [12,13]. Small for gestational age, especially, is a high risk for hyperthyrotropinemia [13]. The incidence of low birthweight was higher among infants with high TSH levels on NBS in our study than among the general population. However, all NBS-positive low-birthweight infants were able to discontinue LT4 replacement by the age of 15 years. Because high TSH levels in low-birthweight infants may be transient, these patients should be excluded when determining the prevalence of permanent CH.

Although this was not a nationwide survey, this is the first study to clarify the true prevalence of permanent CH in Japan. The study was limited by the inability to ascertain the LT4 replacement status in patients transferred to other hospitals, and the lack of investigation of the presence of CH patients who were not detected by NBS. We also did not examine factors associated with transient CH in this study. However, this study was strengthened by uniform data, as a single institution managed almost all the CH screenings in the Niigata Prefecture, and transient CH was excluded from long-term follow-up until the age of 15 years.

## 5. Conclusions

In our study, 62.5% of the LT4 replacement patients discontinued treatment by 15 years of age. From these results, the prevalence of permanent CH in the Niigata Prefecture during this period was 1 in 2500–3500 children.

## Figures and Tables

**Figure 1 IJNS-07-00027-f001:**
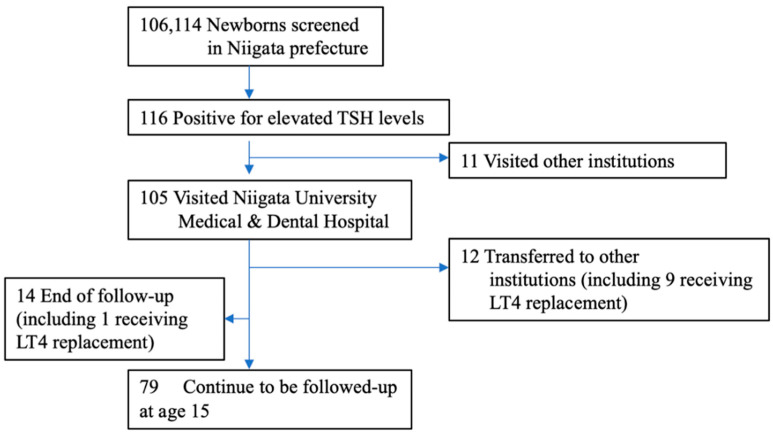
Enrollment of the study subjects. Between April 2002 and March 2006, a total of 106,114 newborns were screened for CH in Niigata prefecture, and 116 were referred to pediatric endocrinologists. We evaluated 105 subjects (90.5%). Eleven patients did not visit our hospital due to reasons such as hospitalization in the neonatal intensive care unit. Fourteen patients (including 1 patient who died after cardiac surgery while on LT4 replacement) with normal thyroid function at the first visit, transient hypothyroidism, or transient hyperthyrotropinemia did not receive further follow-up. Twelve patients, including 9 on LT4 replacement, were transferred to another hospital. At the age of 15, 79 patients were being followed up.

**Figure 2 IJNS-07-00027-f002:**
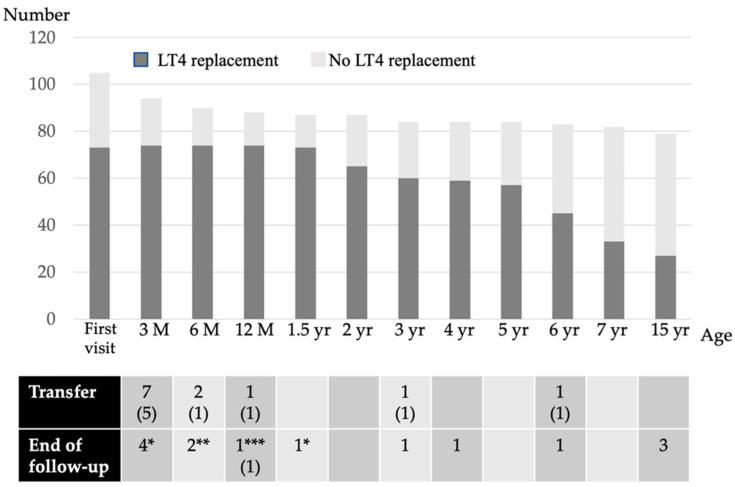
The LT4 replacement status of each age group. M, month; yr, years; (*n*), the number in both parenthesis indicates the number of patients receiving LT4 replacement; * transient hyperthyrotropinemia due to maternal antithyroid drugs or blocking TSH receptor antibody; ** thyroid function was normal from the first visit and the follow-up ended; *** the patient died after cardiac surgery.

**Table 1 IJNS-07-00027-t001:** Subject backgrounds.

Sex (*n*)	Male (54), Female (62)
NBS-positive timing (*n*)	First examination (28), second examination (88)
Whole blood TSH level in filter paper at NBS **	10.5 (9.8–31.3) mIU/L
Birth weight (BW); mean ± SD (range) BW < 2500 g; *n* (%) BW < 1500 g: *n* (%)	2830 ± 664 g (424–3916) 18 (15.5%) 8 (6.9%)
Thyroid morphology * (*n*)	Ectopic (7), hypoplasia (10), enlarged (7), eutopic (81)

*n*, number of patients; NBS, newborn screening; * Data from 105 patients who underwent detailed examinations at our hospital; ** Data are shown as median (interquartile range).

## Data Availability

The data presented are available upon request from the corresponding author. The data are not publicly available because of privacy restrictions.

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
