# Peer review of "Re-Evaluation of the Prevalence of Permanent Congenital Hypothyroidism in Niigata, Japan: A Retrospective Study"

_2409-515X, 2021, doi:10.3390/ijns7020027_

Round 1

Reviewer 1 Report

The manuscript describes the long-term follow-up of newborns who had elevated thyroid stimulating hormone at birth between April 2002 ad March 2006 in Niigata Prefecture, Japan. Some children were followed until 15 years of age following multiple evaluations to determine whether CH was permanent or transient in order to determine the prevalence of permanent CH. Long-term follow-up of CH patients is extremely important and interesting in order to assist with determining cut-offs for newborn screening in addition for determining the incidence of disease. The manuscript should be expanded by delving into TSH values at birth and the TSH cut-offs selected by the NBS program, rather than just concentrating on incidence of disease.

The TSH values at birth should be reported. Were the newborns with the lowest TSH values, the ones with the transient CH? What were the TSH ranges of newborns with permanent CH? Should TSH cut-offs be re-evaluated?

It would be helpful to report the numbers of newborns with ectopic gland or thyroid agenesis amongst the 105 subjects. How many newborns had eutopic thyroid gland (line 68)?

Were there children who were evaluated between 2-5 years of age and required medication and were again evaluated at 15 years of age and no longer required medication? Presumably when they were re-evaluated, it was determined that treatment was needed. There should be a discussion regarding recommendation for treating transient CH.

Some of the writing is confusing and needs to be clarified:

Line 22 of abstract - Exactly 105 subjects presented to our department for the first time. It is not clear what is meant for the first time? At what age were they seen? Were the newborns referred to your center?

Lines 88 and 89. It is not clear what the outcome section refers to?

Figure 3 – some boxes need clarification. E.g. 79 continue to be followed-up at age 15

Line 26 – should this be 62.5% instead of approximately 60%.

The NBS numbers should be reported in section 2.1. How many repeat specimens were requested? How many specimens had a TSH over 30 mU/L and how many were between 8 and 30 mU/L?

Table1. The birth weight are not aligned correctly in the table.

Lines 100, 101 – 32 were left untreated; 10 of the latter had persistently mildly elevated TSH levels and were initiated on LT4 by the age of 1 year. If 10 of the 32 were initiated on LT4, they can’t be described as untreated. Perhaps “initially untreated”?

Line 110 . We evaluated 105 subjects (90.5%) them. Delete “them”.

Transient CH in low birth weight/immature newborns has been reported previously. This should be discussed and references given (line 163-168). [e.g. Grob et al., J Pediatr Endocrinol Metab 2020; 33:375 amongst others]

The discussion section can be improved by including a discussion of long-term follow-up of CH cases in other programs.

Line 152 - Why is the incidence of CH increased from first visit after birth (1 in 1300) to first year of age (1 in 1200)? This needs to be explained.

Author Response

[Responses to the comments of Reviewer 1]

We wish to express our appreciation to the Reviewer for his or her insightful comments.

Comments to the Author:

  • Long-term follow-up of CH patients is extremely important and interesting in order to assist with determining cut-offs for newborn screening in addition for determining the incidence of disease. The manuscript should be expanded by delving into TSH values at birth and the TSH cut-offs selected by the NBS program, rather than just concentrating on incidence of disease.

The TSH values at birth should be reported. Were the newborns with the lowest TSH values, the ones with the transient CH? What were the TSH ranges of newborns with permanent CH? Should TSH cut-offs be re-evaluated?

  • Response: Thank you for your valuable comment. Unfortunately, the TSH level at birth is not measured in Japan, so I cannot provide data. As for the filter paper blood cut-off TSH value in newborn screening, some patients with permanent CH are just above the TSH cut-off value, and some patients with transient CH exceed the filter paper blood TSH value of 30 mIU/L. In our data, I think it is challenging to mention the TSH cut-off value. I would like to focus on the topic of this paper on the incidence of CH. As for your proposal, I would like to consider presenting it as another paper.
  • It would be helpful to report the numbers of newborns with ectopic gland or thyroid agenesis amongst the 105 subjects. How many newborns had eutopic thyroid gland (line 68)?
  • Response: As for the thyroid morphology of the 105 patients, it is shown in Table 1 as patient background.
  • Were there children who were evaluated between 2-5 years of age and required medication and were again evaluated at 15 years of age and no longer required medication? Presumably when they were re-evaluated, it was determined that treatment was needed. There should be a discussion regarding recommendation for treating transient CH.
  • Response: Thank you for your comment. As shown in Figure 2, there are several cases in which LT4 replacement could not be discontinued by the age of 5 years re-evaluation, but could be discontinued at the age og 15 years re-evaluation. We have added to the discussion that even in cases where LT4 replacement is being used as CH, transient CH may be included (L167-169).
  • Line 22 of abstract - Exactly 105 subjects presented to our department for the first time. It is not clear what is meant for the first time? At what age were they seen? Were the newborns referred to your center?
  • Response: 105 newborns have been referred to our center for further examination. We have amended the text of L22-23.
  • Lines 88 and 89. It is not clear what the outcome section refers to?
  • Response: Thank you for your comment. We have clarified our primary and secondary outcomes (L89-91).
  • Figure 3 – some boxes need clarification. E.g. 79 continue to be followed-up at age 15
  • Response: Thank you for your suggestion. We have amended, according to your suggestion.
  • Line 26 – should this be 62.5% instead of approximately 60%.
  • Response: Thank you for your suggestion. We have amended, according to your suggestion (L26 and L187)
  • The NBS numbers should be reported in section 2.1. How many repeat specimens were requested? How many specimens had a TSH over 30 mU/L and how many were between 8 and 30 mU/L?
  • Response: As for the number of NBS in the subject period, we moved it to 2.1 as indicated. Regarding the timing of NBS positive among the subjects, the results are shown in Table 1. Twenty-eight subjects had filter paper blood TSH 30 mIU/L or higher, and 88 subjects had filter paper blood TSH 8 to 30 mIU/L.
  • The birth weight are not aligned correctly in the table.
  • Response: We have amended the description of body weight in Table 1.
  • Lines 100, 101 – 32 were left untreated; 10 of the latter had persistently mildly elevated TSH levels and were initiated on LT4 by the age of 1 year. If 10 of the 32 were initiated on LT4, they can’t be described as untreated. Perhaps “initially untreated”?
  • Response: We have amended, according to your suggestion (L101).
  • Line 110 . We evaluated 105 subjects (90.5%) them. Delete “them”.
  • Response: We have amended, according to your suggestion (L111).
  • Transient CH in low birth weight/immature newborns has been reported previously. This should be discussed and references given (line 163-168). [e.g. Grob et al., J Pediatr Endocrinol Metab 2020; 33:375 amongst others]
  • Response: We added the references for L171-172. We also added that SGA is a high-risk factor for thyrotropinemia.
  • The discussion section can be improved by including a discussion of long-term follow-up of CH cases in other programs.
  • Response: Thank you for your valuable comment. As for the reports on the incidence of CH by long-term follow-up, we have listed references 9 and 10, and we have added reference 11 (L163 and L164). The other paper on long-term follow-up of CH was not accepted because it was a paper on prognosis including intellectual development of CH, which is different from the purpose of this paper.
  • Line 152 - Why is the incidence of CH increased from first visit after birth (1 in 1300) to first year of age (1 in 1200)? This needs to be explained.
  • Response: In this study, the prevalence of CH at 1 year of age has increased from the first visit due to the inclusion of patients who started LT4 replacement in early infancy without LT4 replacement at the initial diagnosis, i.e., persistent mild hyperthyrotropinemia (L154-156).

We believe that the findings of this study are significant and valuable. Thank you once again for your consideration of our paper.

Reviewer 2 Report

The study on the prevalence of permanent CH in Niigata provides valuable long-term follow-up information about newborn screening for CH. The authors demonstrate that the prevalence of treated CH varies during childhood according to age, and that within their population only 40% of children who received thyroxine treatment for primary CH during childhood remained on this at full final height. The clinical follow-up data comes from a single hospital where most children with CH in the region receive care and gives a comprehensive explanation of the criteria used to both start and stop thyroxine treatment.

Limitations –

The major limitation of the methodology used to calculate population prevalence is the inability to capture follow-up details for babies/children who may have received care at another institution. The authors have accounted for this by reducing the population base to 91% and by providing a range of prevalence for both transient and permanent CH. Can the authors provide some information about the hospital and likelihood that children with permanent or transient CH could be cared for at other institutions? In addition, can the authors comment on the presence of permanent or transient CH cases that were missed by screening. If present, these should be added to the calculations of population prevalence.

Minor issues

Lines 61-63. Do 30 and 7 days refer to baby’s age or time from notification/sample collection. Please clarify.

Lines 157-158. The meaning of this sentence is unclear. Is the incidence of transient CH in the US derived from 1:50,000 figure combined with 1:2,500 incidence CH reported in ref 8, and so does “5-10% of NBS-positive children” mean 5-10% NBS-positive children with CH? It may be simpler to refer to an estimated incidence of 50,000 transient CH. Similarly, the author has written that “40-53% NBS-positive children actually have transient CH” – does this refer to 40-53% of screen-detected newborns with CH?

Lines 166-168 – Suggest that the authors clarify that they are talking about determining the incidence of permanent rather than all (permanent and transient) CH.

Author Response

[Responses to the comments of Reviewer 2]

We wish to express our appreciation to the Editor for his or her insightful comments, which have helped us significantly improve the paper.

Comments to the Author:

  • The major limitation of the methodology used to calculate population prevalence is the inability to capture follow-up details for babies/children who may have received care at another institution. The authors have accounted for this by reducing the population base to 91% and by providing a range of prevalence for both transient and permanent CH. Can the authors provide some information about the hospital and likelihood that children with permanent or transient CH could be cared for at other institutions? In addition, can the authors comment on the presence of permanent or transient CH cases that were missed by screening. If present, these should be added to the calculations of population prevalence.
  • Response: Thank you for your valuable comment. Since we do not know the LT4 replacement status of patients transferred to other hospitals, it is listed in L179-180 as limitations. In addition, we have not been able to investigate the CH who were not found in the NBS, so we have added them to the limitation (L180-181)
  • Lines 61-63. Do 30 and 7 days refer to baby’s age or time from notification/sample collection. Please clarify.
  • Response: I added "of birth" because it means the age of the day (L59-61).
  • Lines 157-158. The meaning of this sentence is unclear. Is the incidence of transient CH in the US derived from 1:50,000 figure combined with 1:2,500 incidence CH reported in ref 8, and so does “5-10% of NBS-positive children” mean 5-10% NBS-positive children with CH? It may be simpler to refer to an estimated incidence of 50,000 transient CH. Similarly, the author has written that “40-53% NBS-positive children actually have transient CH” – does this refer to 40-53% of screen-detected newborns with CH?
  • Response: Thank you for your essential remarks. I have corrected NBS-positive children to NBS-positive children with CH (L161 and L162).
  • Lines 166-168 – Suggest that the authors clarify that they are talking about determining the incidence of permanent rather than all (permanent and transient) CH.

Response: Thank you for your comment. As you mentioned, I think it should be limited to permanent CH incidence, so I added "permanent (L176).

Thank you again for your comments on our paper. We trust that the revised manuscript became a better and suitable one for publication. 

Round 2

Reviewer 1 Report

There is some conflicting information from the authors regarding the NBS TSH data that needs to be resolved. The manuscript states (lines 52-54):

NBS Method in the Niigata Prefecture

Blood samples were collected on filter paper within the first 4 to 7 postnatal days, and the TSH level in the filter paper sample was measured using an enzyme-linked immunosorbent assay

However, the authors state: Response: Thank you for your valuable comment. Unfortunately, the TSH level at birth is not measured in Japan, so I cannot provide data

Please clarify the discrepancy. The TSH values at birth, if available, would be useful to include in the manuscript.

In the discussion section there should be a mention of what the recommendations are for treating patients with transient CH since this is a controversial issue.

Line 51 – “In this study, patients on LT4 replacement were defined as CH.” Patients cannot be defined as a disease.

Table 1. “NB-positive timing”. Please expand NB in the figure legend

Author Response

[Responses to the comments of Reviewer 1]

We wish to express our appreciation to the Reviewer for his or her insightful comments.

Comments to the Author:

  • There is some conflicting information from the authors regarding the NBS TSH data that needs to be resolved. The manuscript states (lines 52-54):

NBS Method in the Niigata Prefecture

Blood samples were collected on filter paper within the first 4 to 7 postnatal days, and the TSH level in the filter paper sample was measured using an enzyme-linked immunosorbent assay

However, the authors state: Response: Thank you for your valuable comment. Unfortunately, the TSH level at birth is not measured in Japan, so I cannot provide data

Please clarify the discrepancy. The TSH values at birth, if available, would be useful to include in the manuscript.

  • Response: I assumed that the "TSH value at birth" was the TSH data from the day of birth. The TSH levels at newborn screening have been added to Table 1.
  • In the discussion section there should be a mention of what the recommendations are for treating patients with transient CH since this is a controversial issue.
  • Response: Thank you for your comment. I believe there is no argument against LT4 replacement during a period of hypothyroidism, even if it is transient. For example, DUOX2 abnormality, which is common in Asian races, is a known cause of transient CH, but the timing of improvement in thyroid function varies and may be post-pubertal. Therefore, we should not assume that our patients have permanent CH and should evaluate them several times to confirm whether they are transient CH or not. These are mentioned in the Discussion (L167-169).
  • Line 51 - "In this study, patients on LT4 replacement were defined as CH." Patients cannot be defined as a disease.
  • Response: We have amended the problem you mentioned (L51).
  • Table 1. "NB-positive timing". Please expand NB in the figure legend.
  • Response: NB was a mistake for NBS. It has been corrected and added to the Table footnote.

We believe that the findings of this study are significant and valuable. Thank you once again for your consideration of our paper.
